# Augmenting transcriptome assembly by combining *de novo* and genome-guided tools

Prachi Jain[1,3], Neeraja M. Krishnan[1,3] and Binay Panda[1,2]

[1] Ganit Labs, Bio-IT Centre, Institute of Bioinformatics and Applied Biotechnology, Bangalore, India
[2] Strand Life Sciences, Hebbal, Bangalore, India
[3] These authors contributed equally to this work.

## ABSTRACT

Researchers interested in studying and constructing transcriptomes, especially for non-model species, face the conundrum of choosing from a number of available *de novo* and genome-guided assemblers. None of the popular assembly tools in use today achieve requisite sensitivity, specificity or recovery of full-length transcripts on their own. Here, we present a comprehensive comparative study of the performance of various assemblers. Additionally, we present an approach to combinatorially augment transciptome assembly by using both *de novo* and genome-guided tools. In our study, we obtained the best recovery and most full-length transcripts with Trinity and TopHat1-Cufflinks, respectively. The sensitivity of the assembly and isoform recovery was superior, without compromising much on the specificity, when transcripts from Trinity were augmented with those from TopHat1-Cufflinks.

## INTRODUCTION

High-throughput technology has changed our understanding of many facets of biology, like diseases (*Shendure & Lieberman Aiden, 2012*), plant genetics (*Egan, Schlueter & Spooner, 2012*; *Simon et al., 2009*), and synthetic biology (*Mitchell, 2011*). The advent of DNA microarrays in the 90's ushered an era of high-throughput genome-wide gene expression profiling studies (*DeRisi et al., 1996*; *Golub et al., 1999*; *Schena et al., 1995*; *Schena et al., 1996*). DNA microarray, although a powerful technique, is dependent on gene annotation, and, therefore, the genome sequence information. This is circumvented by RNA sequencing (RNA-seq), which uses next-generation sequencing instruments, and can be leveraged even in the absence of a genome to study the transcripts and their expression. This shift from the semi-quantitative, hybridization-based approaches, as in DNA microarrays, to the quantitative, sequencing-based approaches has tremendously facilitated gene expression analysis. RNA-seq experiments yield additional information on transcriptome characterization and quantification, including strand-specificity, mapping of fusion transcripts, small RNA identification and alternate splicing (*Martin & Wang, 2011*;

Corresponding author
Binay Panda, binay@ganitlabs.in

*Mortazavi et al., 2008*; *Ozsolak & Milos, 2011*; *Waern, Nagalakshmi & Snyder, 2011*; *Wang, Gerstein & Snyder, 2009*).

A number of tools have been developed for transcriptome assembly. The process of assembly is quite complex. Factors such as varying expression levels among genes, presence of homologues and spliced isoforms are responsible for this. The complexity is handled by assemblers differently. Studies have identified *k*-mer size (*Gruenheit et al., 2012*; *Robertson et al., 2010*) and seed length for read alignment (*Schatz, Delcher & Salzberg, 2010*) as important parameters for short-read transcriptome assembly. The optimal *k*-mer size for an assembly was found to be positively correlated with the number of reads used during assembly (*Zerbino & Birney, 2008*). Experimental evidence also supports the importance of *k*-mer length in distinguishing low- from highly-expressed genes during assembly (*Gibbons et al., 2009*). Genome-guided tools use the seed length parameter while aligning reads to a reference genome prior to assembly. An optimal seed length depends on the quality of the reference genome and is set to a smaller value for divergent genomes (*Darling et al., 2004*). Varying seed length varies the speed versus accuracy tradeoff during the process of assembly.

Transcriptome assemblers may be classified as *de novo* or genome-guided tools. In our study, we used Trinity (*Grabherr et al., 2011*; *Haas et al., 2013*), SOAPdenovo-Trans (*Xie et al., 2013*), Oases (*Schulz et al., 2012*) and Trans-ABySS (*Robertson et al., 2010*) from the *de novo* tools category, and TopHat-Cufflinks, (*Kim et al., 2013*; *Roberts et al., 2011*; *Trapnell, Pachter & Salzberg, 2009*; *Trapnell et al., 2012*; *Trapnell et al., 2010*) and Genome-guided Trinity (*Haas, 2012*) under the second category. *De novo* assembly tools create short contigs from overlapping reads, which could be extended based on insert size estimates. They could either involve construction of *de Bruijn* graphs using *k*-mers (for short reads) or using an overlap-layout-consensus approach (for longer reads) (*Nagarajan & Pop, 2013*). TopHat-Cufflinks is an alignment-cum-assembly pipeline, that involves spliced read alignment to the genome and their subsequent assembly into transcripts. It uses the genome template and read pairing to produce full-length or near full-length transcripts. In genome-guided Trinity, the reads are aligned to the genome and partitioned into read clusters, which are then individually assembled using Trinity.

Both *de novo* and genome-guided approaches have their own advantages (*Martin & Wang, 2011*). *De novo* assembly tools are independent of the genome sequence, alignment of reads to the genome and, thus, the ambiguity underlying this process. They can recover transcript fragments from regions missing in the genome assembly. On the other hand, genome-guided assembly approaches are relatively faster, less resource-intensive, can filter out contamination and sequencing artifacts, can recover low-abundance transcripts, and can fill gaps using the genome sequence resulting in full-length transcripts. In the absence of a reference genome, genomes of closely-related organisms can be used for genome-guided transcriptome assembly (*Collins et al., 2008*; *Salzberg et al., 2008*; *Toth et al., 2007*).

Different assemblers have different ranges of sensitivity and specificity, but none of them comes close to assembling all the reads without any errors, into valid transcripts. To enhance the detection sensitivity, one can think of combining assemblies from *de novo* and

genome-guided approaches. However, any approach introduces at least some level of error during assembly. These errors typically include splicing errors, redundancy, chimerism and gaps. Therefore, along with the advantage of enhancement in sensitivity while combining assemblies, there is also a risk of loss of specificity due to compounding of errors.

Here, we describe a detailed comparison of existing *de novo* and genome-guided assemblers, and determine the best combination of those that can be used to augment assembly, while increasing sensitivity and keeping the false assemblies to a minimum. Given the growing importance of RNA-seq, this study will help establish a better workflow for transcriptome assembly.

## MATERIALS AND METHODS

### Simulating RNA-seq reads

The *Arabidopsis thaliana* (TAIR10) complete genome sequence, coordinates for genes and transposons were downloaded from ftp://ftp.arabidopsis.org/home/tair/Genes/TAIR10_genome_release and the GFF file was parsed to obtain exonic coordinates. These were used to simulate Illumina-like RNA-seq reads using Flux-simulator (*Griebel et al., 2012*) (FS-nightly-build_1.1.1-20121119021549) with the following options supplied within a parameter file (Supplemental Methods): NB_MOLECULES (Number of RNA molecules initially in the experiment): 5000000; SIZE_DISTRIBUTION (Gaussian distribution of fragment size as N(Mean, Standard deviation)): N(300,30); READ_NUMBER (Total number of reads to be simulated): 4000000; READ_LENGTH: 76; PAIRED_END: YES; ERR_FILE (Inbuilt Illumina error model for 76nt reads): 76. The resultant Illumina-like reads were split into 2 fastq files corresponding to read1 and read2 using a Python script from the Galaxy tool suite (Supplemental Methods).

### Read assembly

The simulated reads were assembled using four *de novo* (Trinity r2012-06-08; Trans-ABySS v1.3.2; Oases v0.2.08; SOAPdenovo-Trans v1.0) and two genome-guided (TopHat1 v2.0.4 that uses Bowtie1 (*Langmead, 2010*) v0.12.8 and Cufflinks v2.0.0; genome-guided Trinity r2012-10-05 that uses GSNAP (*Wu & Nacu, 2010*) r2012-07-20 [V3]) transcriptome assembly pipelines. Trinity and genome-guided Trinity use a fixed *k*-mer size of 25nt. We used Trans-ABySS on ABySS (*Simpson et al., 2009*) (v1.3.3) multi-*k*-mer assemblies (ranging from 20–64nt), Oases on Velvet (*Zerbino & Birney, 2008*) (v1.2.07) multi-*k*-mer assemblies (every alternate *k*-mer ranging from 19–71nt) and SOAPdenovo-Trans with a fixed *k*-mer size of 23nt. We also tested the TopHat2 (*Kim et al., 2013*) (v2.0.7 that uses Bowtie v2.0.5)-Cufflinks pipeline on the simulated data but did not observe any difference in the assembly statistics compared to the TopHat1-Cufflinks pipeline that uses Bowtie1. All assemblers were run using default parameters (details in Supplemental Methods). However, we fixed the parameter for minimum length of assembled fragment as 76nt (equal to the length of the read). In the case of SOAPdenovo-Trans, contigs were used instead of scaffolds for all downstream analyses, as the minimum length cutoff could not be set for scaffolds.

### Redundancy assessment using CD-HIT-EST

We used CD-HIT-EST (*Li & Godzik, 2002*) (v4.5.4) to assess redundancy in each assembly. It retained the longest sequence out of a cluster of sequences that share at least 95% sequence similarity (−c 0.95) on either strand (−r 1), based on a word size of 8 (−n 8). The accurate but slow mode (−g 1) was used for clustering. Further details on other optional parameters used to run CD-HIT-EST are provided in Supplemental Methods.

### Model assembly

We defined all read-covered transcript regions (TAIR10) as Model Assembly or MA as previously described in the report by *Mundry et al. (2012)*.

### Calculation of N50 and $N_{(MA)}50$ statistics for an assembly

N50 is defined as the minimum contig length for 50% of the assembly after sorting the contigs in the descending order of their lengths. We calculated N50 values for all assemblies, MA and the TAIR10 simulated transcripts, pre- and post-CD-HIT-EST. We also calculated the N50 values for each assembler, while taking the MA cumulative size (nt) as the denominator instead of the respective assembly sizes. We termed these N50 values as the $N_{(MA)}50$.

### Mapping assemblies to MA using Megablast

The assembled fragments (query) were mapped against the MA fragments (subject) using Megablast (*Altschul et al., 1990*) (blast+ v2.2.26) with default parameters (Supplemental Methods). The Megablast hits were parsed in order to either maximize query coverage (to compute misassembly statistics) or maximize subject coverage (to compute MA recovery statistics). This was done to choose the best hits and discard the partially overlapping hits when the unique coverage was lower than or equal to 10nt. The scripts used to remove nested and partial overlaps from Megablast hits are provided in Supplemental Methods. For misassembly statistics, each assembled fragment was categorized, based on mapping, to either belong to a single MA source ($\geq$90% mapping) or be chimeric across multiple sources (misassembled, <90% mapping).

### Expression level bins

We estimated the average "per-nucleotide coverage" (pnc) for all MA fragments, based on their read support, as given below.

$$\frac{(\text{No. of Reads} \times \text{Read Length})}{\text{MA fragment length}}.$$

The MA fragments were then categorized into 8 expression level bins, B1 to B8, the pnc for each being: 1 for B1, $>1$ & $\leq$2 for B2, $>2$ & $\leq$3 for B3, $>3$ & $\leq$4 for B4, $>4$ & $\leq$5 for B5, $>5$ & $\leq$10 for B6, $>10$ & $\leq$30 for B7, and $>30$ for B8. We chose denser sampling for the lower pnc values and sparser sampling for the higher pnc values since we observed the distribution of MA fragments to be denser in the lower pnc categories (Fig. S1).

### Recovery of isoforms

Using simulated data, we obtained the MA-equivalent for the exonic regions of isoform-bearing genes, and performed a Megablast search of the assemblies against it (Supplemental Methods). The Megablast hits were parsed to maximize subject coverage, after removing nested and partial overlaps (same as described earlier). For all assemblers, we calculated the number of exons recovered per isoform and the length recovery of each exon.

### Augmenting Trinity assembly with TopHat1-Cufflinks assembly

We mapped the Trinity assemblies against the TopHat1-Cufflinks transcripts using Megablast. Using the subtractBed option from BEDTools (*Quinlan & Hall, 2010*), the regions unique to TopHat1-Cufflinks transcripts were identified. These unique transcript regions were used to augment the Trinity assembly.

### Receiver Operating Characteristic (ROC) Curve

The ROC curve was plotted for each assembler using the simulated data. The sensitivity (True Positive Rate, TPR) was estimated as % total length recovered by each assembler out of the total MA size. The FPR (False Positive Rate, 100-specificity) was estimated as the % assembled fragments that did not map to the MA fragments. For TopHat1-Cufflinks assembly, and the unique regions to TopHat1-Cufflinks used to augment Trinity, mapping was performed against TAIR10 transcripts.

### Zebrafish transcriptome data analyses

To exemplify the recovery trends of isoforms and non-isoforms, we assembled the transcriptome of zebrafish, *Danio rerio*, embryo (2dpf) using an RNA-seq dataset from SRA (ERR003998, containing 22,286,504 paired reads, 76nt in length with a 200bp insert size). These reads were assembled using *de novo* and genome-guided transcriptome assemblers in the same manner as described for the simulated data (Trinity r2012-06-08; Trans-ABySS v1.3.2; Oases v0.2.08; SOAPdenovo-Trans v1.02; TopHat1 v2.0.4 that uses Bowtie1 v0.12.8 and Cufflinks v2.0.0; genome-guided Trinity r2012-10-05 that uses GSNAP r2013-03-31 [V5]). The ENSEMBL zebrafish genome, Zv9, was used as the reference for genome-guided assemblies. Further, 85 transcripts from the zebrafish hox gene cluster were downloaded from ENSEMBL. These correspond to 49 genes, 23 of which harbour 59 isoforms. We focused on the recovery of these transcripts by all assemblers.

Since isoforms carry greater sub-sequence similarity than non-isoforms, we pooled all the 85 transcripts and identified their shared and unique regions using Megablast. RNA-seq reads were then mapped to these, again using Megablast, in order to identify their MA-equivalents and to estimate their average pnc. This served as the reference for recovery comparison across all assemblers. Finally, the recovery of MAs in the shared and unique categories was estimated also using Megablast, as described earlier, while maximising subject coverage.

## RESULTS

We compared the performance of assemblers using a variety of assembly-based parameters (numbers and lengths of assembled fragments, N50, $N_{(MA)}50$ and extent of redundancy) and mapping-based parameters (recovery of MA fragments (numbers and lengths), mis-assembly, reliance on pnc) for isoforms and non-isoforms, and shared and unique transcript regions.

### Assembly statistics

We simulated Illumina-like paired-end 76nt RNA-seq reads for the exonic regions of the *Arabidopsis thaliana* TAIR10 genome (see Methods for details), covering 15,532 TAIR10 transcripts. The transcript regions contiguously covered by reads were termed as Model Assembly (MA) fragments and were used as a valid reference for mapping the assemblies. A given transcript, therefore, comprised of one or more MA fragments, the shortest being 76nt in length. We obtained 70,382 MA fragments from the 15,532 TAIR10 transcripts.

The reads were assembled using Trinity (TNY), Trans-ABySS (TA), Oases (OS), SOAPdenovo-Trans (SDT), TopHat1-Cufflinks (TC) and genome-guided Trinity (GGT) (see Methods for details) with default parameters and a minimum assembled fragment length of 76nt. We compared the numbers of assembled fragments, the minimum and maximum assembled fragment lengths, their length frequency distributions, the N50 and $N_{(MA)}50$ statistics for the six assemblers, pre- and post-redundancy assessment in the assembly using CD-HIT-EST.

The total number and lengths of assembled fragments varied widely across the six assemblers (Table 1). The shortest fragment reported by all assemblers was 76nt (as fixed by the minimum reporting length threshold). The range of fragments at the long end varied across assemblers with the longest for TopHat1-Cufflinks at 10,502 nt (same as the maximum MA fragment length, Table 1). This was expected since TopHat1-Cufflinks is a genome-guided assembler that allows recovery of full-length or near full-length transcripts.

We observed a 4- and ~1.5-fold reduction in assembly size post-CD-HIT-EST for Trans-ABySS and Oases, respectively (Table 1). For Trinity, genome-guided Trinity and SOAPdenovo-Trans, we observed no difference pre- and post-CD-HIT-EST across the entire distribution of assembled fragment lengths (Fig. 1). However, Trans-ABySS resulted in a higher number of longer, redundant assembled fragments (Fig. 1) and a low number of shorter assembled fragments.

Like the results from the frequency distribution statistics for long- and short-assembled fragments, Trinity, genome-guided Trinity and SOAPdenovo-Trans yielded N50 values closer to that of MA. However, Trans-ABySS, Oases and TopHat1-Cufflinks yielded N50 numbers that were much higher than that of the MA. Post-CD-HIT-EST, the N50 values were not affected for most assemblers except for Trans-ABySS (Table 1).

In contrast, the $N_{(MA)}50$ values, which are not dependent on the total size of any assembly, were lower than that of the MA for Trinity, genome-guided Trinity and SOAPdenovo-Trans, both pre- and post-CD-HIT-EST. For Trans-ABySS and Oases, the

**Table 1  Assembly statistics pre- and post-CD-HIT-EST.**

| | TAIR10 transcripts | Model Assembly (MA) | Trinity | Trans-ABySS | Oases | SOAP denovo-Trans | Tophat1-Cufflinks | Genome guided Trinity |
|---|---|---|---|---|---|---|---|---|
| **Pre CD-HIT-EST** | | | | | | | | |
| Total transcripts or fragments | 15532 | 70382 | 45978 | 65594 | 52533 | 39533 | 8671 | 46064 |
| Transcriptome size (nt) | 2.61E+07 | 1.67E+07 | 1.27E+07 | 6.62E+07 | 4.21E+07 | 1.02E+07 | 1.15E+07 | 1.27E+07 |
| N50 (nt) | 1952 | 720 | 684 | 1390 | 1453 | 617 | 1624 | 646 |
| N(MA)50 (nt) | 1952 | 720 | 264 | 2802 | 2519 | 114 | 1136 | 254 |
| Median transcript/fragment length (nt) | 1456 | 85 | 115 | 804 | 523 | 103 | 1124 | 121 |
| Min. transcript/fragment length (nt) | 201 | 76 | 76 | 76 | 76 | 76 | 76 | 76 |
| Max. transcript/fragment length (nt) | 14623 | 10501 | 6888 | 7326 | 8128 | 6300 | 10502 | 7833 |
| **Post CD-HIT-EST** | | | | | | | | |
| Total transcripts or fragments | 14615 | 62528 | 45281 | 16060 | 28051 | 39471 | 8395 | 42968 |
| Transcriptome size (nt) | 2.45E+07 | 1.56E+07 | 1.21E+07 | 1.15E+07 | 1.85E+07 | 1.02E+07 | 1.11E+07 | 1.20E+07 |
| N50 (nt) | 1951 | 788 | 627 | 1194 | 1485 | 617 | 1621 | 661 |
| N(MA)50 (nt) | 1951 | 788 | 260 | 798 | 1680 | 155 | 1197 | 273 |
| Median transcript/fragment length (nt) | 1451 | 87 | 113 | 477 | 216 | 103 | 1121 | 120 |
| Min. transcript/fragment length (nt) | 201 | 76 | 76 | 76 | 76 | 76 | 76 | 76 |
| Max. transcript/fragment length (nt) | 14623 | 10501 | 6888 | 7326 | 8128 | 6300 | 10502 | 7833 |

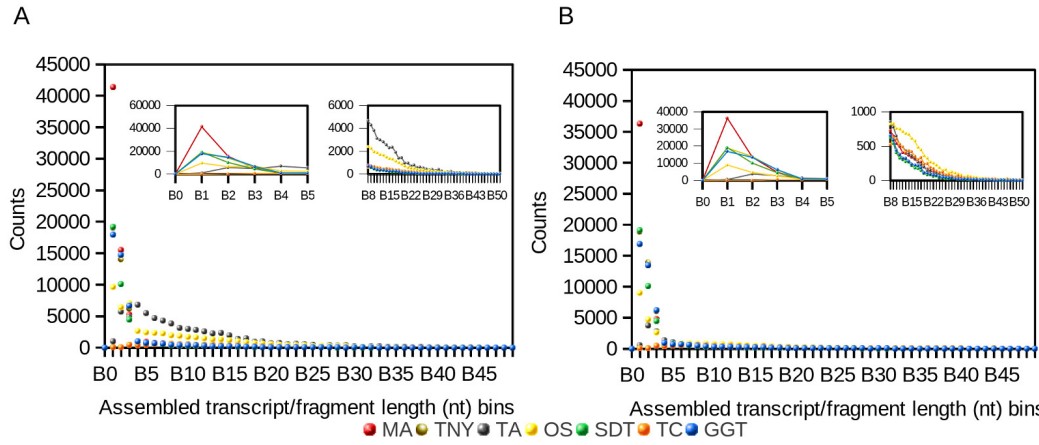

**Figure 1  Frequency distribution of lengths (nt) of Model Assembly (MA) & assembled transcript fragments before (A) and after (B) CD-HIT-EST using simulated data.** X-axis values are assembled fragment length ranges in nucleotides (nt) and Y-axis values are the numbers of assembled fragments in each length bin (Counts). TNY, Trinity; TA, TransABySS; OS, Oases; SDT, SOAPdenovo-Trans; TC, TopHat1-Cufflinks; GGT, genome-guided Trinity.

$N_{(MA)}50$ values were relatively higher, and were reduced ~3 and ~1.5-fold respectively, post-CD-HIT-EST. The $N_{(MA)}50$ values were higher than that of the MA for TopHat1-Cufflinks, both pre- and post-CD-HIT-EST.

## Mapping-based statistics

MA fragments, from TAIR10 transcripts corresponding to transposable elements and transcript isoforms as per the GFF annotation, were not included in the expression-based binning. We excluded these elements since they have stretches of identical sequences, which posed a problem in assigning the pnc index due to unreliable mapping of reads and assemblies to the correct isoform and/or transposable element. Out of a total of 70,382 MA fragments, we were thus left with 46,805 MA fragments, which still had certain level of sub-sequence similarity, presumably arising from gene paralogs and SSRs. The numbers of MA fragments finally obtained in different expression bins were: 15,706 in B1 (the lowest expression bin), 9,722 in B2, 5,205 in B3, 2,797 in B4, 1,820 in B5, 4,522 in B6, 4,862 in B7 and 2,171 in B8 (the highest expression bin).

## MA recovery statistics

The assembled transcript fragments from all assemblers were mapped to the MA using Megablast to assign common identifiers for comparisons. We compared the numbers and lengths of MA fragments recovered within all expression bins, B1 to B8, using the best hits from Megablast.

Trinity detected the highest number of MA fragments in B1, followed by genome-guided Trinity and SOAPdenovo-Trans (Fig. 2). However, the number of unique transcripts detected was highest for TopHat1-Cufflinks, followed by Trinity, for the lowest expression bin B1 (Fig. 2). The extent of overlapping (common detections) between Trinity and either genome-guided Trinity or SOAPdenovo-Trans was 30% for the lowest expression bin B1. The number of MAs detected was highest for Trinity for all expression bins (B1–B8), with other assemblers showing better recovery with increase in expression. In comparison to Trinity, the detection sensitivity for Oases and Trans-ABySS increased with increased expression, as stated above, but the overall number of recovered MA fragments remained lower in all expression bins (Fig. 2). TopHat1-Cufflinks showed a drop in its detection sensitivity from B1 to B2, and a steady rise thereafter. For all assemblers, with the exception of Trans-ABySS, we observed an increase in higher-order intersections (common detections by 2, 3, 4, 5 or all 6 assemblers) across bins B2–B8, proportional to the decrease in unique detections and lower order intersections (Fig. 2).

We observed that longer MA fragments tend to have higher read support, thus justifying their assignment only to the higher expression bins, B3 onwards (Fig. 3, Fig. S1). The shorter MA fragments displayed all levels of read support, and, therefore, were assigned to all expression bins, B1 to B8 (Fig. 3, Fig. S1). Given this non-uniform distribution of lengths in each expression bin, we compared the length recovery across assemblers, for each expression bin, in different MA fragment length categories (Fig. 3).

For the shortest length category, 76nt, the median length recovery was close to 100% across all expression bins for all assemblers, with the exception of Trans-ABySS (Fig. 3). The outlier (datapoints falling outside the interquartile range of % MA length recovery) trend was highly variable across assemblers, and clustered closer to the median for SOAPdenovo-Trans, followed by Trinity and genome-guided Trinity. For Trinity,

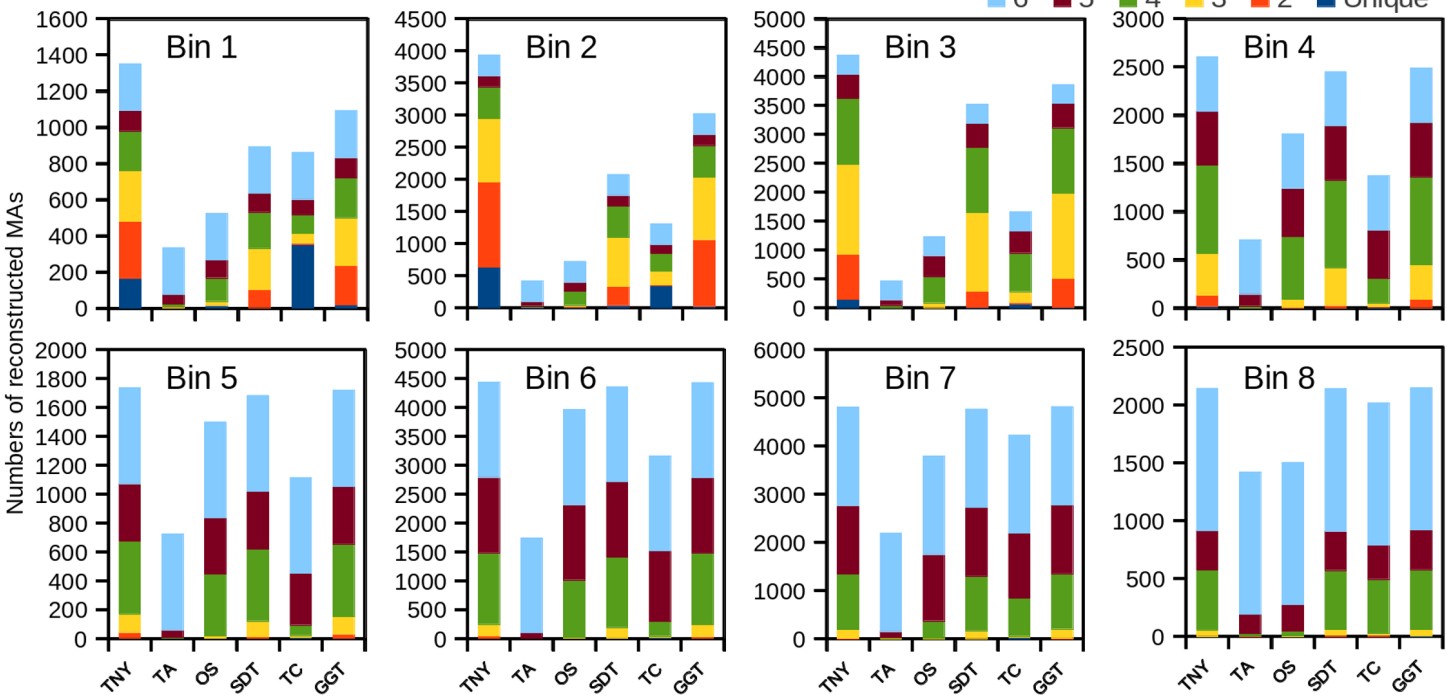

**Figure 2** **The number of recovered Model Assembly (MA) fragments in different expression bins (Bin 1–8) using simulated data.** The recovery of MA fragments by 6 different assemblers was estimated across eight expression level categories (B1–B8) and then binned by per-nucleotide coverage (pnc). The recovered fragments are presented as unique to an assembler or as an overlap between 2, 3, 4, 5, or all the 6 assemblers for each expression bin. TNY, Trinity; TA, TransABySS; OS, Oases; SDT, SOAPdenovo-Trans; TC, TopHat1-Cufflinks; GGT, genome-guided Trinity.

Trans-ABySS, Oases and genome-guided Trinity, we observed the outliers clustering around ∼40% length recovery (∼30nt) for the MA fragments in the lowest expression bin (B1) that matched the word size for Megablast (28nt). The outliers in the lowest expression bin for TopHat1-Cufflinks, and those in the highest expression bin for Oases, spanned all the way from ∼40% to the median.

For the subsequent length categories, we observed a gradual increase in the median length recovery from as low as 20% to all the way up to 100%, with an increase in the expression levels. This gradual increase was also seen, to a minor extent, for the same bin across MA length categories for most assemblers. TopHat1-Cufflinks showed a median recovery of 100% across all expression bins and all MA fragment length categories. The pattern of length recovery was similar for Trinity, SOAPdenovo-Trans and genome-guided Trinity across all the expression bins and MA fragment length categories. Overall, these three assemblers outperformed Trans-ABySS and Oases in terms of higher median MA fragment lengh recovery and tighter distribution around the median (Fig. 3).

## Misassembly statistics

We used Megablast-based mapping to evaluate the accuracy of assembled fragments, and determine whether each assembled fragment belonged to a single MA source or was a chimera across multiple sources (misassembled). We classified the assembled

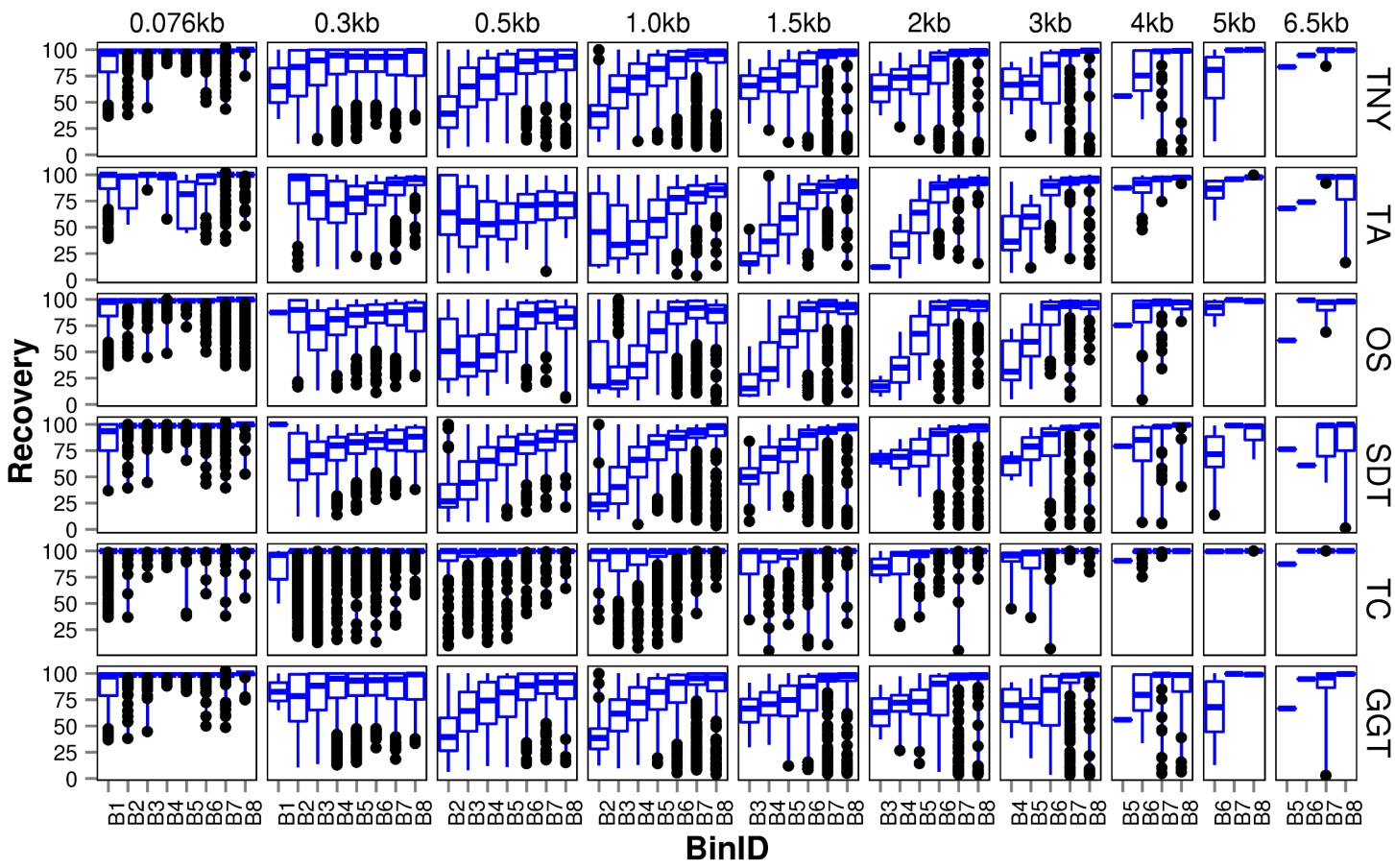

**Figure 3 Recovered length (%) of Model Assembly (MA) fragments using simulated data.** MA fragments were binned into different length categories (≤76b, 76–300b, 300–500b, 500b–1kb, 1–1.5kb, 1.5–2kb, 2–3kb, 3–4kb, 4–5kb, 5–6.5kb), and the length recovered for each assembler across expression bins (B1–B8) was visualized separately for each MA length bin. The black dots represent the outliers, the boxes represent the 25%–75% (Q1–Q3) interquartile range (IQR), the middle lines in the boxes represent the median, and the blue solid lines represent the whiskers from these boxes till the minimum/maximum of the length range. Outliers fall below Q1 − (1.5 × IQR) or above Q3 + (1.5 × IQR). TNY, Trinity; TA, TransABySS; OS, Oases; SDT, SOAPdenovo-Trans; TC, TopHat1-Cufflinks; GGT, genome-guided Trinity.

fragments into three categories, ≥90%, between 60%–90%, and <60%, based on the extent of their lengths mapped to any single MA fragment. They were further classified into various assembled fragment length categories to check their relationship with assembly quality, pre- and post-CD-HIT-EST. Trans-ABySS and Oases showed relatively higher numbers of misassembled fragments (between 60%–90%, and <60% mapping) in the 200–400nt and 500–600nt ranges, respectively (Fig. 4). In the case of Trans-ABySS, the extent of misassembly decreased post-CD-HIT-EST, whereas in the case of Oases, it went up. We did not observe any difference in the extent of misassembly, pre- and post-CD-HIT-EST, for any other assembler (Fig. 4). The extent of misassembly was overestimated in TopHat1-Cufflinks as it additionally assembled transcript regions not represented by reads, and therefore the MA, using the genomic information (Table 1). For Trinity, SOAPdenovo-Trans and genome-guided Trinity, there was a clear trend of decreasing misassembly with increasing assembled fragment length (Fig. 4). The degree of
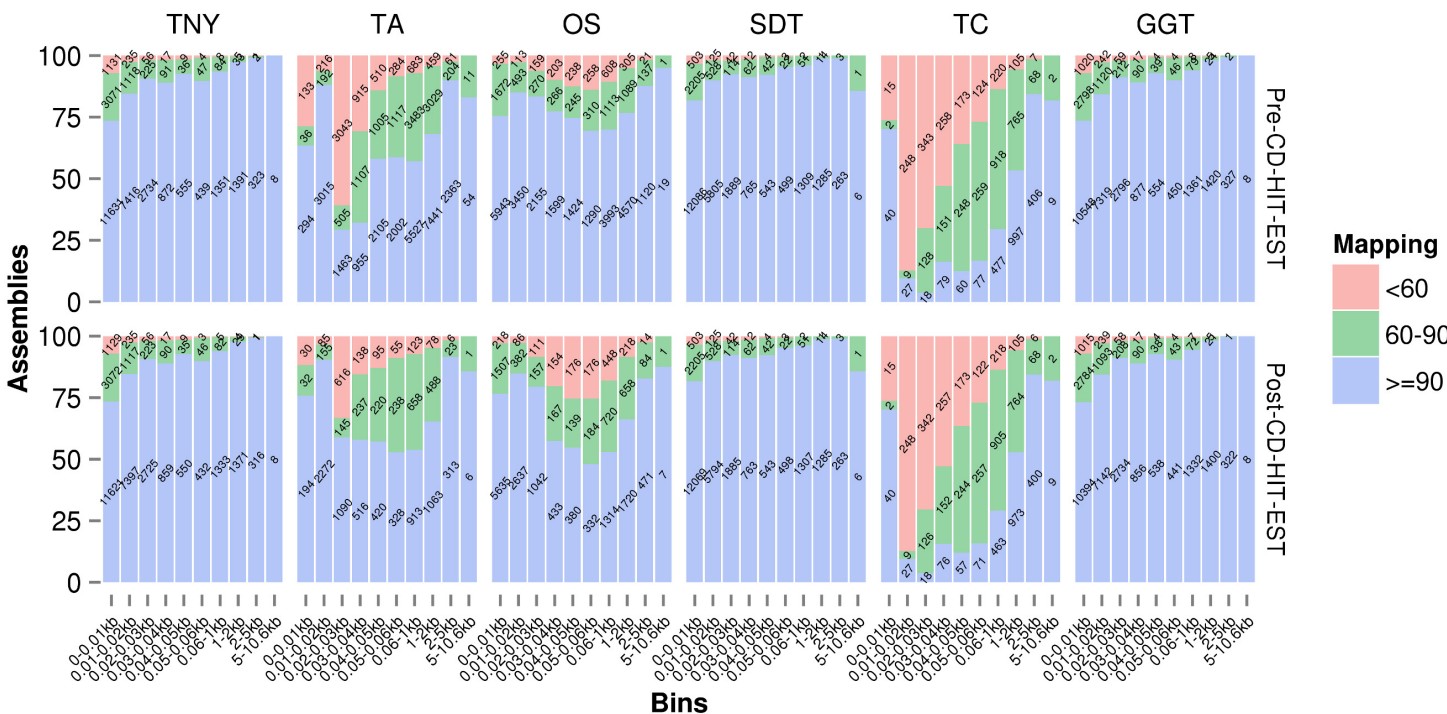

**Figure 4 Assembly mapping statistics using simulated data.** The assembled fragments were classified as >90%, between 60%–90% and <60% categories based on the fraction mapped to a single Model Assembly (MA) fragment, pre- (A) and post- (B) CD-HIT-EST. TNY, Trinity; TA, TransABySS; OS, Oases; SDT, SOAPdenovo-Trans; TC, TopHat1-Cufflinks; GGT, genome-guided Trinity.

misassembly was less for SOAPdenovo-Trans than for Trinity or genome-guided Trinity for the shorter fragments.

## Recovery of isoforms using simulated and zebrafish data

Next, we compared the extent of recovery of isoforms from the transcripts assembled with various assemblers. We had a total of 24,846 exons in the MA fragments corresponding to 2,970 isoforms. We mapped assembled fragments from all assemblers to these exons, using Megablast, and maximized the exon coverage. Trinity, followed by genome-guided Trinity and SOAPdenovo-Trans recovered the highest numbers of isoforms to 80%–100% of their length (Fig. 5). We observed a correlation between the median pnc for each length recovery category and the numbers of exons recovered per isoform. The median pnc was in the range of 2–3 for unrecovered isoforms and 17–23 for those which were fully recovered or close to fully recovered by all assemblers. Trans-ABySS and Oases, which showed a relatively lower recovery, correlated with a higher median pnc, suggesting a higher threshold of pnc needed for good recovery by those assemblers (Fig. 5).

In order to test whether our understanding of isoform recovery from simulated reads holds true for a real dataset, we measured the recovery of shared and unique transcript regions of the hox gene cluster in zebrafish (see Methods for details). We observed that across assemblers, Trinity, followed by genome-guided Trinity and SOAPdenovo-Trans recovered most and closer to full-length transcripts (Figs. 6–8). TopHat1-Cufflinks

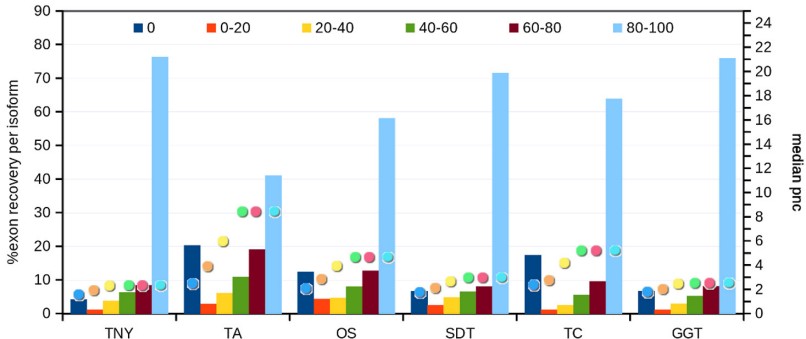

**Figure 5 Isoform recovery statistics using simulated data.** Recovery was estimated for the number of exons per isoform in different Model Assembly (MA) length recovery categories (0%, >0%–20%, >20%–40%, >40%–60%, >60%–80% and >80%–100%). The median per-nucleotide coverage (pnc) of isoforms for each of these categories, represented as dots, was also estimated for each assembler. TNY, Trinity; TA, TransABySS; OS, Oases; SDT, SOAPdenovo-Trans; TC, TopHat1-Cufflinks; GGT, genome-guided Trinity.

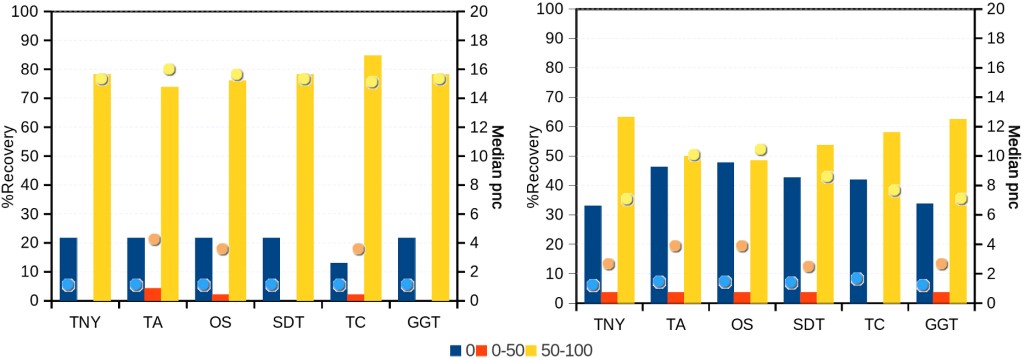

**Figure 6 Recovery of transcripts in the unique and shared regions of hox gene cluster in zebrafish.** The unique and shared regions of hox gene cluster from zebrafish *Danio rerio* with a minimum per-nucleotide coverage (pnc) of 1 were extracted. The recovery was measured for these regions in the 0%, 0%–50% and 50%–100% recovery categories. The median pnc of isoforms for each of these categories is represented as dots. TNY, Trinity; TA, TransABySS; OS, Oases; SDT, SOAPdenovo-Trans; TC, TopHat1-Cufflinks; GGT, genome-guided Trinity.

recovered mostly full-length, but fewer, transcripts. Trans-ABySS and Oases recovered fewer and truncated transcripts. The length recovery (% of MA length) by all assemblers, except TopHat1-Cufflinks, displayed a dependency on pnc which was more obvious in the recovery of shared regions of transcripts, as expected due to a wider range of read depth (Figs. 6–7).

## Augmenting transcriptome assembly

Since each assembler produced a set of unique transcripts or fragments, we proceeded towards augmenting the assemblies, one with another. We ruled out most combinations of assemblers, and chose Trinity and TopHat1-Cufflinks, as they produced the maximum number of valid transcript fragments and full-length transcripts, respectively. After aligning the Trinity transcript fragments to the TopHat1-Cufflinks transcripts and

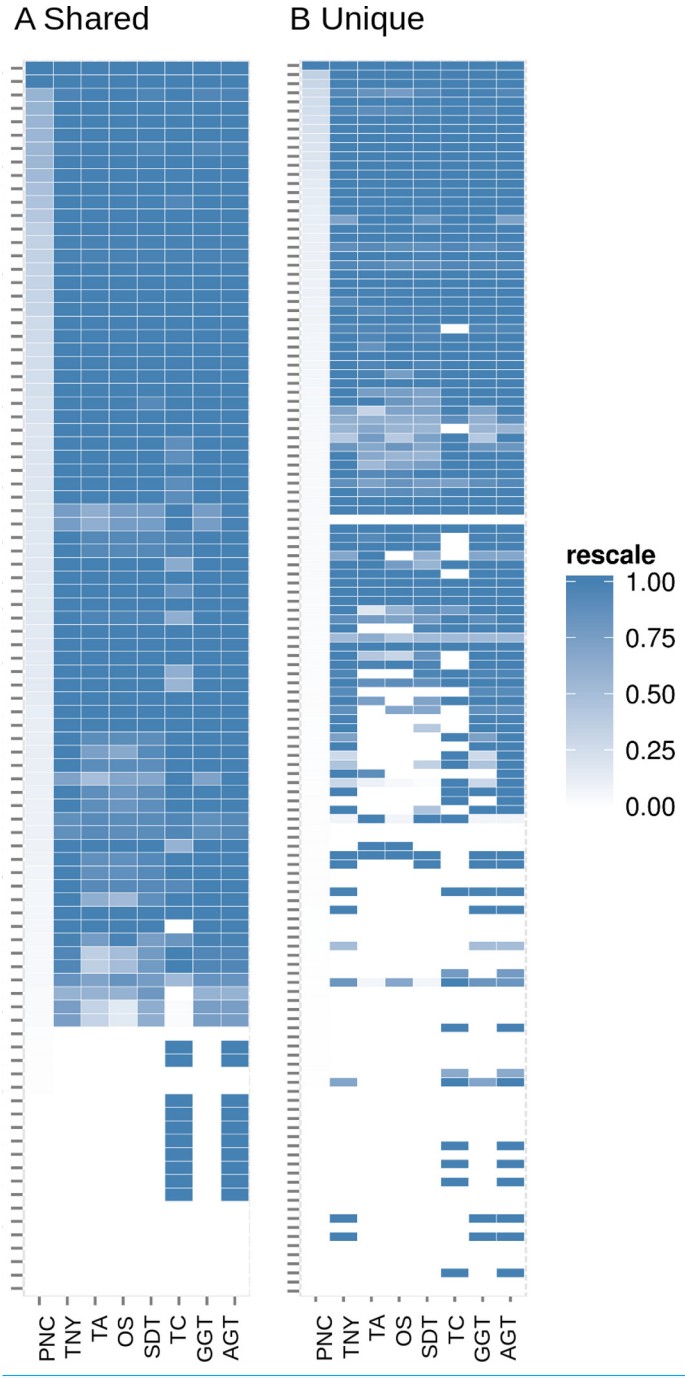

**Figure 7 Heatmap analyses of %length recovery of shared and unique transcript regions of the zebrafish hox gene cluster.** The transcript regions were ranked in a descending order of their pncs. TNY, Trinity; TA, TransABySS; OS, Oases; SDT, SOAPdenovo-Trans; TC, TopHat1-Cufflinks; GGT, genome-guided Trinity; AGT, Augmented Trinity.

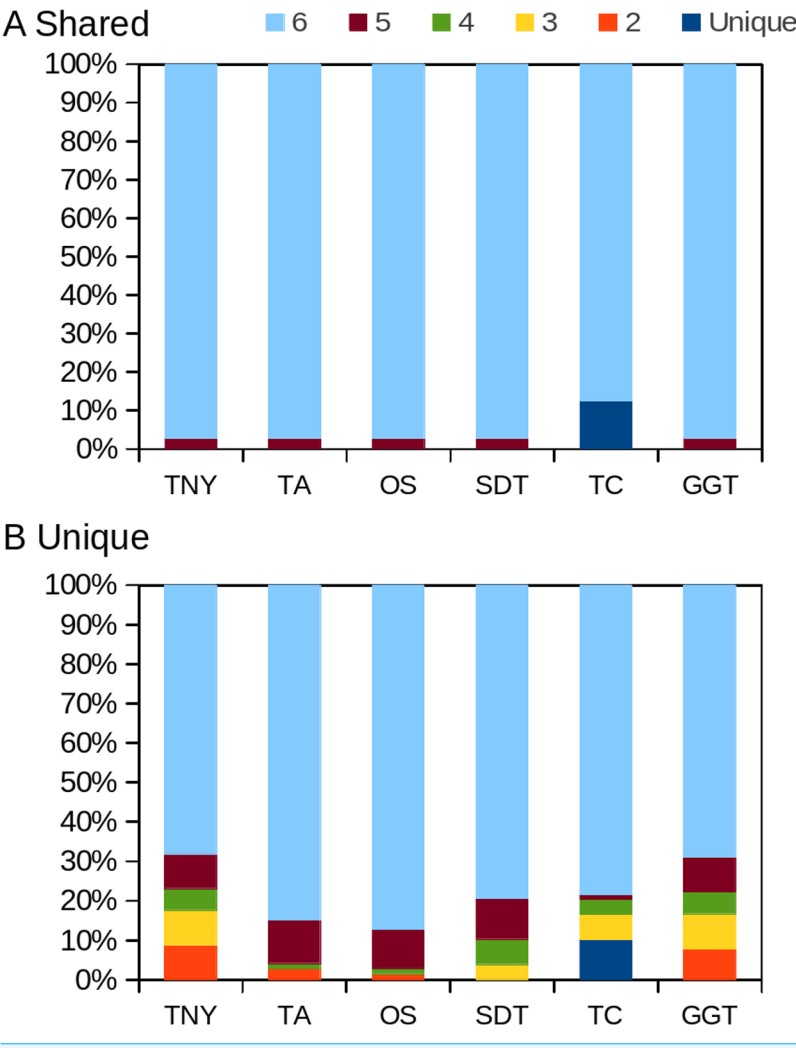

**Figure 8 Intersection histogram of recovered transcripts from the shared and unique regions of the zebrafish hox gene cluster.** TNY, Trinity; TA, TransABySS; OS, Oases; SDT, SOAPdenovo-Trans; TC, TopHat1-Cufflinks; GGT, genome-guided Trinity.

augmenting the unique regions from TopHat1-Cufflinks to the Trinity assembly, we obtained a cumulative size increase of 1.37 Mb in the assembly size (∼20% of original Trinity assembly). The augmented assembly detected 1,377 MAs more than the original Trinity assembly, and the MA recovery increased by 5,23,127nt after augmentation. Relaxing the stringency of Megablast word size, from 28 to lower, would have improved the MA length recovery further, though at the expense of losing isoform reconstruction capability and sensitivity to variation in read-depth maintained by the fragmented structure of Trinity transcriptome assembly. We further observed that the median length recovery for the augmented Trinity (AGT) was overall better than Trinity alone (Fig. 9). For MAs longer than 1500nt, this augmentation yielded full-length transcripts across all expression levels (Fig. 9). Even for MAs around 500nt in length, we saw the advantage of augmentation, at least for recovering more number of full-length transcripts. The outliers

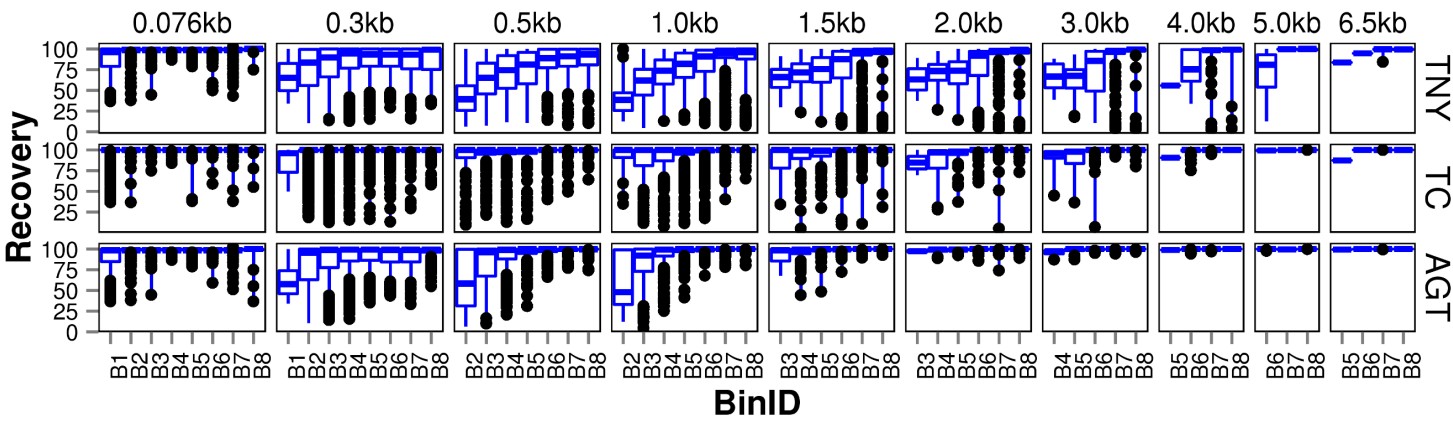

**Figure 9 Recovered length (%) of fragments after augmenting Trinity-derived transcripts with that from TopHat1-Cufflinks using simulated data.** The length recovery was visualized in the same manner as described in Fig. 3. TNY, Trinity; TC, TopHat1-Cufflinks; AGT, Augmented Trinity.

in augmented Trinity were fewer than TopHat1-Cufflinks in all length categories. The improved recovery with augmented Trinity proves that an integrative approach is useful, particularly when one has access to genome in addition to the RNA-seq reads.

## DISCUSSION

RNA-seq using next-generation sequencing is a powerful technology to understand the transcriptome of an organism. Although, genome-guided assemblers like TopHat-Cufflinks can assemble full-length transcripts, most *de novo* approaches and even genome-guided Trinity, where the genome is used only to partition the RNA-seq reads, are useful in detecting novel transcripts. In our comparative study, we found that each assembler produced a set of unique transcripts or fragments, especially at lower levels of expression (Fig. 2). Therefore, we started by asking whether one can obtain a better transcriptome assembly by augmenting a *de novo* assembly with that from a genome-guided approach. While this was a reasonable question to ask, the presence of multiple tools, and the errors associated with each of them, compounded the problem. We started by finding out the efficiency of individual tools, the ones that are popularly used by the community, with the hope that we could choose from assemblers and combine the results from them without compounding additional errors in the final assembly.

A lower threshold for minimum reported fragment length allows one to retain valid assemblies (as demonstrated in Fig. 3) at the expense of increasing errors. However, in our observations with read length as the minimum reported fragment length, the errors were comparatively lesser than the number of valid assemblies (Fig. 4). Regardless of the minimum reported fragment length, the extent of misassembly was different for different assemblers. Oases and Trans-ABySS resulted in more misassemblies than the other tools (Fig. 4). We suspected that a large number of redundant transcripts produced by Oases and Trans-ABySS were possibly misassembled. Interestingly, however, when we compared the misassembly statistics pre- and post-CD-HIT-EST, we found that Oases misassembled more frequently in the non-redundant regions, compared to Trans-ABySS (Fig. 4).

Due to higher sub-sequence similarity in isoforms, which contain more shared regions than non-isoforms, the chances of misassembly are greater. The shared regions among transcripts also pose additional difficulty in discerning their true source at the time of estimating recovery using mapping. We analyzed the shared and unique regions within isoforms separately in order to distinguish their recovery and underlying pnc patterns.

We mapped all the reads back to the model assemblies (MA) for measuring per-nucleotide coverage (pnc). Given that the read-tracking is absent during the process of assembly, we chose to not assign pnc to individual assemblies, since re-mapping of reads using aligners might result in multiple hits. This compels the user to arbitrarily assign the pnc to an assembled transcript fragment, which may or may not be the same as the pnc estimated by an assembler. With simulated reads, the actual (simulated) and estimated pnc (using Megablast) of MAs were found to be correlated (Fig. S2). Based on this, we expect the estimated pnc of transcript fragments by any assembler to be positively correlated with their actual pnc.

In addition to the simulated dataset, we used the zebrafish hox gene cluster for transcript recovery analysis. Vertebrate hox genes are known to be involved during development, are arranged in sets of uninterrupted clusters, and are in most cases expressed in a collinear fashion (*Kuraku & Meyer, 2009*). Since they include both isoforms and non-isoforms, they were ideal candidates for our comparative analyses on recovery of shared and unique regions. We assembled RNA-seq reads from zebrafish embryo and identified the fragments related to the hox gene cluster transcripts. We expected the shared regions to have a greater read depth than the unique regions. Indeed, we found this to be reflected in terms of a greater median pnc for the shared regions in the $>50\%$ recovery category (Figs. 6 and 7).

N50 is an assembly attribute widely used to compare the quality of genome and transcriptome assemblies. In the absence of the knowledge of actual length distribution in the sequenced dataset (transcripts sequenced), it is generally assumed that a higher N50 implies a better assembly. We know that errors and redundancy in the transcriptome assembly affect its total size. We observed that the range of assembled fragment lengths was variable across assemblers (e.g., TopHat1-Cufflinks produced more full-length transcripts and Trinity produced more number of small fragments than the rest of the assemblers). Therefore, basing a quality metric on the total assembly size and assembled fragment lengths to benchmark the assembly, like what the N50 does, can be highly inaccurate. Instead, a quality metric based on the expected size of the transcriptome, NT50, or on the read-covered regions of the transcriptome, $N_{(MA)}50$, tends to provide a more accurate benchmark. Indeed, we found that the $N_{(MA)}50$ accurately reflects the recovery quotient of an assembler (Table 1). The reduction in $N_{(MA)}50$ pre- and post-CD-HIT-EST is most reflective of the reduction in the assembly numbers (Table 1). In addition to $N_{(MA)}50$, we found that the median, interquartile range, minimum, maximum and outliers for transcript assembly length were more useful in describing a transcriptome assembly.

Misassembly can occur as a result of subsequence similarity within reads which manifests as highly branched nodes in a *de Bruijn* graph. This subsequence similarity, along with mismatches/errors in sequencing reads, can also cause spurious blast hits that

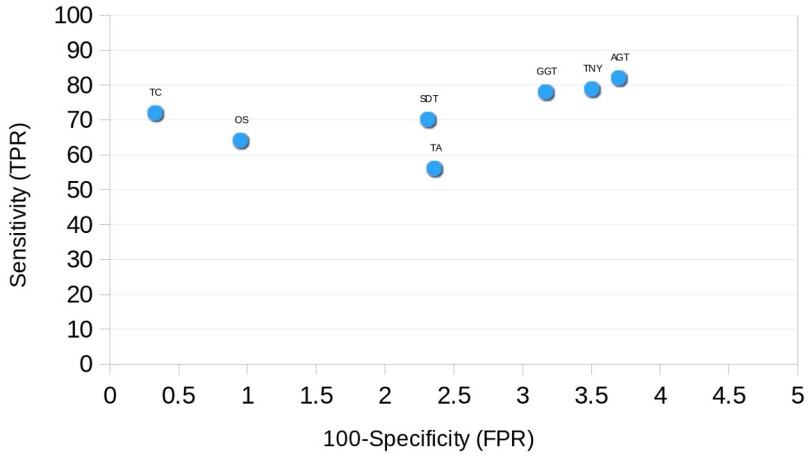

**Figure 10  Receiver Operating Characteristic (ROC) curve for transcriptome assemblers using simulated data.** The sensitivity (True Positive Rate, TPR) and 100 - specificity (False Positive Rate, FPR) were estimated as % total length recovered for each assembler out of the total Model Assembly (MA) size and as the % assembled fragments that did not map to the MA fragments respectively. For TopHat1-Cufflinks assembler, and the unique regions to TopHat1-Cufflinks used to augment Trinity, the mapping was performed with the *A. thaliana* TAIR10 transcripts. TNY, Trinity; TA, TransABySS; OS, Oases; SDT, SOAPdenovo-Trans; TC, TopHat1-Cufflinks; GGT, genome-guided Trinity; AGT, Augmented Trinity.

are seen as outliers in the box and whisker plot for MA recovery (Fig. 3). Spurious blast hits resulting from subsequence similarity span from Megablast word size of 28nt onwards. We observed fewer outliers of this kind in multi-*k*-mer based approaches like Oases and Trans-ABySS. SOAPdenovo-Trans also appears to contain fewer spurious blast hits, especially for the 76nt MA length category. It is known that the SOAPdenovo algorithm discards a repetitive node if there is unequal read support on edges to and from that node, and only builds parallel paths carrying the node if there are equal number of reads on either edges (*Li et al., 2010*). Based on our observation of low redundancy in assemblies in the 76nt MA length category (Fig. 3), and higher % of assemblies mapping correctly in the lower assembled fragment length categories (Fig. 4), we postulate that SOAPdenovo-Trans discarded repetitive sequences from the assembly for these length categories. TransABySS, pre-CD-HIT-EST, showed a higher frequency of assembled fragment lengths >200nt (Fig. 1A). The frequency of assembled fragments in this length range was much lower, post-CD-HIT-EST (Fig. 1B), suggesting higher redundancy in this range.

Finally, we found Trinity to perform best in transcript recovery across all expression levels (Figs. 2 and 3), and the median length recovery by TopHat1-Cufflinks to always be ∼100% (Fig. 3). These findings held true even with varied lengths and sequencing coverages of RNA-seq reads (Table S1). Hence, we chose to combine the Trinity assemblies with those from TopHat1-Cufflinks. This resulted in the detection of ∼1300 more transcript fragments, corresponding to a cumulative increase in recovery of nearly 0.5Mb. The process resulted in an augmented assembly with greater sensitivity and only minimal compromise in its specificity, while maintaining the expression-based fragmented assembly structure of Trinity (Fig. 10). This proved that an integrative approach may be

employed for recovering more transcript fragments, particularly when one has access to genome assembly in addition to RNA-seq reads.

## ACKNOWLEDGEMENTS

We thank Professor N. Yathindra for encouragement.

### Funding

This research is funded by the Department of Electronics and Information Technology, Government of India (Ref No: 18(4)/2010-E-Infra., 31-03-2010) and the Department of Information Technology, Biotechnology & Science and Technology, Government of Karnataka (Ref No: 3451-00-090-2-22). Both the grants were received by the Bio-IT Centre. The funders had no role in study design, data collection and analysis, decision to publish, or preparation of the manuscript.

### Grant Disclosures

The following grant information was disclosed by the authors:
Department of Electronics and Information Technology, Government of India: Ref No: 18(4)/2010-E-Infra., 31-03-2010.
Department of Information Technology, Biotechnology & Science and Technology, Government of Karnataka: Ref No: 3451-00-090-2-22.

### Competing Interests

Prachi Jain and Neeraja M. Krishnan are employed by Ganit Labs. Binay Panda is employed by Ganit Labs and Strand Life Sciences.

### Author Contributions

- Prachi Jain and Neeraja M. Krishnan performed the experiments, analyzed the data, contributed reagents/materials/analysis tools, wrote the paper.
- Binay Panda conceived and designed the experiments, wrote the paper.

### Supplemental Information

Supplemental information for this article can be found online at http://dx.doi.org/10.7717/peerj.133.

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
