# Peer review of "Augmenting transcriptome assembly by combining de novo and genome-guided tools"

_PeerJ, doi:10.7717/peerj.133_

## Round 0.1 · original submission · Major Revisions

Two reviewers have now provided comments on your paper. There are a number of issues that need to be addressed, including justification for many of the choices made in your simulation studies and better descriptions of figures (including fixes in the figures like axis labels etc.).

·

Basic reporting

Abstract: A sentence or two describing the general findings (which assembly tools performed best, which augmentation produced the best results given your measures, etc.) should be included.


Figures in general need a bit of work. Many of the figures do not have properly labeled axes or the labels are hard to read. Also, many of the legends need to be more explicit about what is contained within the figure. The following are all places where figures should be improved:

Figure 1 – I don’t think the axes are labeled correctly – x-axis should have unit (bp) included; is the y-axis actual frequencies or just counts?

Figure 2 – y axis label? Category labels on x-axis impossible to read.

Figure 4 – very hard to read

Figure 5 – be more explicit about the fact that the points are the median pnc value

Figure 6 – again, legend needs explanation of dots on figures.

Figure 10 – define ROC, TPR, and FPR.

Experimental design

The experimental design is appropriate for the research described.

Validity of the findings

No comments.

Reviewer 2 ·

Basic reporting

- Title: instead of “Augmenting transcriptome assembly combinatorially”, something like “Improving transcriptome assembly by combining de novo and reference-based tools” would be clearer and more informative.

- Abstract is too general. All but the last sentence is background. Shorten the background part in abstract and give more specific information on methods and main conclusions.

- Page 1 line 28: either include the SOAPdenovo-Trans webpage or cite the publication instead of citing SOAPdenovo2. http://arxiv.org/pdf/1305.6760v1.pdf. SOAPdenovo-Trans and SOAPdenovo2 are quite different from each other and were developed by different people.

- Page 2 line 52: “Both de novo and genome-guided approaches have their own strengths”. What are the known strengths of both approaches before this study?

- Page 2 line 54: “However, the process of combining assemblies from multiple assemblers is not error free and may produce false assemblies”. Are there any previously investigated methods and known problems of merging results from multiple tools?

Experimental design

- Page 2 line 70: Current day Illumina reads are usually either 50bp or 100bp. Why picking 76bp? Why not testing multiple lengths? Why picking a specific read number instead of testing several read numbers? Why using simulation instead of real data? I am not suggesting that you should do all the possible tests but it would help to add a few sentences to justify your choice of parameters and datasets.

- Page 2 line 70: explain what these parameters mean so that people who are not familiar with Flux-simulator can understand them. For example, what does “ERR_FILE 76” mean?

- Page 3 line 75: why choosing this particular dataset for analysis? You explained it in the discussion but such information should be moved to either introduction or methods. In addition, give some basic details on the dataset instead of just direct the readers to a table in the original publication. Are the reads paired end or single end? How long are the reads? What is the insertion size? How many reads are there?

- Page 3 line 99: you can filter the size of scaffolds using a simple python or perl script. The contigs do not use any paired end information and are very different from scaffolds.

- Page 2 line 110: explain the important settings in the main text so that the readers do not have to flip back and forth to get the essential information.

- Page 2 line 126: did you use the assembly before or after cd-hit-est? Specify it.

- Page 2 line 129: what criteria did you use to identify misassemblies from megablast results? State it in the main text.

- Page 3 line 148: Why only test isoforms on simulated data instead of both datasets?

- Page 3 line 148: did you use the assembly before or after cd-hit-est? Specify it.

- Methods should include combining multiple tools. How did you combined the assemblies?

- How did you calculate the sensitivity and specificity for the ROC curve?

- Page 8 line 359: What does it mean by “expressed in a collinear fashion”, temporarily collinear or co-expressed as an operon? Why being “collinear” makes it an ideal test for assembly? I have a hard time following the logic flow here.

Validity of the findings

- Figure 1 is difficult to read. The points are too big and overlap each other. Consider re-organize it.

- Figure 2: Legends on the x-axis are too small to read. Tilt them 45 degree and enlarge them.

- Figure 3: What does dividing MAs into size bins tell us? Either discuss the implications more explicitly in the text or combine the bins and make Figure 3 having less panels.

- Figure 4: Numbers on the bars are too small to read.

- Figure 5 legend is confusing. What does it mean by, for example, “>80-100% recovery categories”? Figure legend should be self-contained and shouldn’t require referring to the methods to understand.

- Page 6 line 235: What does the term “outlier” mean? Consider specify it or use a different term.

- All figure legends: Specify which dataset (Arabidopsis vs. Zebra fish) each figure is illustrating

- Page 9 line 403: What kind of relationship did you find, positive or negative? I’m lost here.

- Any thoughts on why combining Trinity and Tophat-Cufflinks turns out to be superior to genome-guided Trinity?

Additional comments

- The manuscript investigates an important aspect of RNA-seq assembly with valid methods.

- Overall, the English language is confusing and doesn’t flow well at places.

- Consider consolidate Figure 3 and 9, and simplify or remove some of the figures. Ten figures, some of which are quite difficult to read, is more than enough for illustrating the message in this paper.

---

## Round 0.2 · accepted · Accept

Thank you for your careful revision and accommodating the reviewers' critiques. I think the paper reads much better now and you have taken care of all their concerns in a productive way.